# Characterization of Hyaluronidase 4 Involved in the Catabolism of Chondroitin Sulfate

**DOI:** 10.3390/molecules27186103

**Published:** 2022-09-18

**Authors:** Shuhei Yamada, Shuji Mizumoto

**Affiliations:** Department of Pathobiochemistry, Faculty of Pharmacy, Meijo University, 150 Yagotoyama, Tempaku-ku, Nagoya 468-8503, Japan

**Keywords:** chondroitin sulfate, glycosaminoglycan, hyaluronidase, hydrolase

## Abstract

Hyaluronidases (HYALs) are endo-beta-*N*-acetylhexosaminidases that depolymerize not only hyaluronan but also chondroitin sulfate (CS) at the initial step of their catabolism. Although HYAL1 hydrolyzes both CS and HA, HYAL4 is a CS-specific endoglycosidase. The substrate specificity of HYAL4 and identification of amino acid residues required for its enzymatic activity have been reported. In this study, we characterized the properties of HYAL4 including the expression levels in various tissues, cellular localization, and effects of its overexpression on intracellular CS catabolism, using cultured cells as well as mouse tissues. *Hyal4* mRNA and HYAL4 protein were demonstrated to be ubiquitously expressed in various organs in the mouse. HYAL4 protein was shown to be present both on cell surfaces as well as in lysosomes of rat skeletal muscle myoblasts, L6 cells. Overexpression of HYAL4 in Chinese hamster ovary cells decreased in the total amount of CS, suggesting its involvement in the cellular catabolism of CS. In conclusion, HYAL4 may be widely distributed and play various biological roles, including the intracellular depolymerization of CS.

## 1. Introduction

Hyaluronidase (HYAL), hyaluronan (HA)-degrading enzyme, was first identified as a “spreading factor” before the discovery of HA [1]. Enzyme activity to hydrolyze HA was named HYAL in the first half of the twentieth century. Degradation of chondroitin sulfate (CS) by HYAL was first reported in 1950 [2]. Because of the structural similarity between HA and CS, HYAL has been considered to be the enzyme responsible for catabolism of not only HA but also CS [3]. HA and CS are both linear polysaccharides composed of a repeating disaccharide unit of [-4-D-glucuronic acid (GlcA)β1-3-*N*-acetyl-D-glucosamine (GlcNAc)β1-]_n_ and [-4-GlcAβ1-3-*N*-acetyl-D-galactosamine (GalNAc)β1-]_n_, respectively. CS is diversely sulfated at hydroxy groups of various positions, although HA is not modified with sulfation at all. CS has various biological and physiological functions, such as cell proliferation, cell migration, cell–cell adhesion, cell–extracellular matrix (ECM) interaction, regulation of cellular signal transduction, and skeletal development [4,5,6]. These activities are considered to be dependent on its specific sulfation modifications.

The major disaccharide units in CS are GlcA-GalNAc(4-*O*-sulfate) (A unit), GlcA-GalNAc(6-*O*-sulfate) (C unit), and nonsulfated GlcA-GalNAc (O unit). However, rare highly sulfated disaccharide units, such as GlcA(2-*O*-sulfate)-GalNAc(6-*O*-sulfate) (D unit) and GlcA-GalNAc(4,6-*O*-disulfate) (E unit), are also found. The special functions of these highly sulfated disaccharide units have been reported. CS chains rich in D or E units were demonstrated to show activity to promote neurite outgrowth of neuronal cells through specific cell surface receptors [7], integrin αVβ3 and contactin-1 [8,9], respectively. CS chains in the secretory granules of mast cells were found to be rich in E units [10]. Highly sulfated CS or CS-dermatan sulfate (DS) hybrid chains have been shown to regulate the cellular signal transduction of various signaling molecules, such as midkine, pleiotrophin, basic fibroblast growth factor, Wnt/β-catenin, semaphoring-3A, and hepatocyte factor [4,11,12,13].

Although the mechanism of CS biosynthesis has been well characterized [14,15], its cellular catabolism is still unclear. HYAL has been considered to hydrolyze CS polysaccharides into small oligosaccharides, followed by degradation with exo-type lysosomal enzymes including β-glucuronidase, β-*N*-acetyl hexosaminidase, and sulfatases in lysosomes. The HYAL family consists of six and seven members in human and rodent genomes, respectively [16,17]. Among HYAL family members, which are homologous in their amino acid sequences, HYAL1 and testicular HYAL (SPAM1) can act on both HA and CS. Although HA has been considered to be the preferred substrate of HYALs, HYAL1 and SPAM1 cleave CS-A more effectively than HA [18]. CS-C and highly sulfated CS are poor substrates for these enzymes, and DS cannot be hydrolyzed at all [19]. However, HYAL4, which is also a member of the HYAL family, was demonstrated to specifically depolymerize CS chains but not act on HA in 2010 [18]. HYAL4 turned out not to be a HYAL but a CS-hydrolase. Characterization of the substrate specificity of HYAL4 showed that it hydrolyzed CS chains preferentially on the saccharide sequence containing highly sulfated disaccharide D units [20,21], suggesting that this enzyme may be involved in regulation of the biological function of highly sulfated CS chains.

HYAL4 is a unique enzyme that exclusively acts on CS and prefers highly sulfated substrates. However, the known characteristics of this enzyme are limited. Previously, the expression of HYAL4 was shown to be restricted to the placenta, testis, and skeletal muscle in humans and testis in mice [20,21,22], suggesting that HYAL4 has specific temporal functions in these organs, such as myogenesis and fertilization. However, recently, its expression in other normal tissues such as mast cells has been reported [23,24]. The typical locations of HYAL4 remain to be established [24]. The optimal pH of HYAL4 was reported to be 4.5–5.0 [20,21], indicating that it is a lysosomal enzyme. However, the C-terminal region of the HYAL4 gene contains the putative glycosyl-phosphatidyl-inositol (GPI)-anchor attachment signal, suggesting that the enzyme may be located in the lipid raft on the cell surface. The cellular localization of HYAL4 is also unclear. Therefore, in this study, we performed characterization of HYAL4 including its cellular localization, assessed the expression pattern of its mRNA and protein in the mouse body, and examined its effects on cellular degradation of CS in cultured cells.

## 2. Results

### 2.1. Expression Analysis of Hyal4 mRNA in Various Tissues

Csóka et al., reported that the expression of human *HYAL4* was restricted to the placenta and skeletal muscle [22]. Using similar primer sets, we also found the restricted expression of human *HYAL4* mRNA in the testis in addition to the placenta and skeletal muscle as well as mouse *Hyal4* mRNA in the testis and 17-day-old embryos. However, CS exists ubiquitously in mammalian tissues, and recently, Farrugia et al. demonstrated that human mast cells produce human HYAL4, which functions in maintaining α-granule homeostasis [23]. Therefore, we re-evaluated the expression pattern of mouse *Hyal4* by PCR using newly designed primers.

The mouse *Hyal4* transcript in multiple tissues was amplified from MTC Multiple Tissue Panels (Clontech) prepared from a BALB/c mouse, and it was ubiquitously expressed (Figure 1), being inconsistent with the previous results. Similar ubiquitous expression was also observed, when cDNAs prepared from various tissues of C57BL/6 and ICR mice were used as the templates (results not shown). Based on these data, mHyal4 may be systemically distributed in mice.

The expression of mouse *Hyal4* mRNA was semi-quantitatively analyzed by real-time RT-PCR to examine where it is predominantly present, using MTC Multiple Tissue Panels as the template (Figure 2). The expression relative to *Gapdh* was high in smooth muscle, the prostate, lymph, placenta, and thymus, although other tissues also showed marked expression of mouse *Hyal4*.

### 2.2. Expression Analysis of Mouse HYAL4 Protein in Various Tissues

The expression level of HYAL4 protein in each organ of the C57BL/6 mouse was investigated by Western blotting after sodium dodecyl sulfate-polyacrylamide gel electrophoresis (SDS-PAGE) under reducing conditions. When a biotinylated anti-HYAL4 (A-7) antibody and a streptavidin–horseradish peroxidase (HRP) conjugate were used, the background level was too high to detect specific protein bands (data not shown). Instead, an HRP-linked anti-HYAL4 (A-7) antibody could successfully stain protein bands (Figure 3), although the background level was still high. This may be due to a mouse IgG antibody raised against mouse HYAL4. When we used rat tissues for Western analysis, the background level was low, and specific bands were clearly detected (Appendix A). The molecular weight of mouse HYAL4 protein can be calculated as ~53 kDa. However, as the band of recombinant mouse HYAL4 expressed in cultured cells was discernible at a molecular mass of 80 kDa [21], mouse HYAL4 must be post-translationally modified, possibly by glycosylation. In this study, the protein band was detected at around 70 kDa in the testis, spleen, lung, and heart as well as at around 135 kDa in the liver by Western blotting analysis. The band detected at around 70 kDa may be a glycosylated form of mouse HYAL4 protein. The band detected at around 135 kDa in the liver sample has not been investigated to determine whether it is also derived from a mouse HYAL4 protein. Bands detected at around 35 kDa may be non-specific, because they were also detected on the control membrane using HRP-linked normal IgG.

### 2.3. Cellular Localization of Mouse HYAL4 Protein

mHyal4 possesses a putative GPI-anchored domain in the C-terminal region, is considered to be a GPI-anchored protein, and may function on the cell surface. However, the optimum pH of mHyal4 is 4.5–5.0, and it may be involved in the catabolism of CS under acidic conditions in an endosome or a lysosome. The cellular localization of HYAL4 was examined using an L6 rat myoblast cell line. The amino acid sequences between mHyal4 and rat Hyal4 show as much as 92% identity, and the anti-HYAL4 (A-7) antibody detects both mHyal4 and rat HYAL4. Flow cytometric analysis of L6 cells was carried out using the anti-HYAL4 (A-7) antibody and a fluorescence-labeled secondary antibody to examine the expression of HYAL4 on the cell surface. As shown in Figure 4, the peak shifted to the higher fluorescence intensity when incubated with the anti-HYAL4 antibody, suggesting the presence of HYAL4 on the cell surface. To confirm the existence of HYAL4 in the cell fraction of L6 cells, analysis using a subcellular proteome extraction kit was carried out. Cells were solubilized and extracted into four fractions containing cytosolic proteins, organelle and membrane proteins, nuclear proteins, and cytoskeletal proteins. Each fraction was subjected to Western blotting, and the results are shown in Figure 5A. The band of HYAL4 was detected in the organelle and membrane proteins fraction, suggesting its distribution in the cell membrane. Although the nuclear fraction also contained the HYAL4 band, it may have overflowed from the organelle and membrane proteins fraction.

Since the optimum pH of mouse HYAL4 is 4.5–5.0, the presence of HYAL4 in the lysosome fraction was also examined. The lysosomal fraction was purified from L6 cells using a commercial Lysosome Enrichment Kit, and the band of HYAL4 was detected in the purified fraction by Western blotting (Figure 5C), indicating that mouse HYAL4 is also present in lysosomes.

Based on the findings, mouse HYAL4 is distributed on cell surfaces as well as in lysosomes.

### 2.4. Effects of Overexpression of Mouse HYAL4 on CS Metabolism of CHO Cells

To investigate the cellular function of HYAL4 as a CS-degrading enzyme, the amount of CS disaccharides was measured before and after overexpression of HYAL4 in Chinese hamster ovary (CHO) cells. The mouse HYAL4 gene was stably expressed in the CHO cell line, although an additional protein band was detected in the overexpressed cells, which has a higher molecular weight than the intact HYAL4 protein (Appendix A). CS and heparan sulfate chains were purified from the Hyal4-overexpressing cells as well as control cells. After enzymatic digestion with bacterial chondroitinase ABC, the digest was derivatized with a fluorophore, 2-aminobenzamide, and analyzed by anion-exchange HPLC on a PA-G column. Although the amount of heparan sulfate chains was not significantly reduced, that of CS from the Hyal4-expressing CHO cells decreased markedly (~27%) compared with the control cells (Table 1), suggesting that HYAL4 most likely hydrolyzes CS chains in the cells and is involved in the cellular metabolism of CS. However, we cannot exclude the possibility that the overexpression of HYAL4 may affect the biosynthetic enzymes of CS to reduce CS production.

## 3. Discussion

In this study, we investigated the cellular localization of HYAL4 to obtain data to elucidate the function of HYAL4 in cells. HYAL4 is an endo-β-*N*-acetylgalactosaminidase that specifically acts on CS, a component ubiquitously present in animal bodies, but the tissue distribution of HYAL4 was reported to be limited. Therefore, we reinvestigated the expression of its mRNA using newly designed primers as well as its protein by Western blotting using various tissues and organs, and clarified the systemic distribution of HYAL4, although the levels differed by tissue. High expression was observed in the prostate or testis, being consistent with previous observations. This suggests that HYAL4 may play more important roles in these organs, such as involvement in male fertilization.

Cellular localization analysis of HYAL4 demonstrated its presence in lysosomes as well as in cell membranes, suggesting that HYAL4 functions both on the cell surface as well as in lysosomes. Since the optimal pH of HYAL4 is acidic (4.4–5.5), it is conceivable that it hydrolyzes CS polysaccharides in lysosomes. However, its role on the cell surface is unclear. HYAL4 has a putative GPI-anchored domain in its C-terminal region, and it may be present in the lipid-raft on the cell surface. Based on accumulating evidence, lipid-rafts play roles in forming platforms that function in membrane signaling and trafficking [25,26]. Involvement of CS-proteoglycan in cellular signal transduction has also been indicated [4,6,27]. HYAL4 on the cell surface may interact with CS without cleaving and regulate its cellular signal transduction.

The critical role of HYAL4 in the cleavage of CS chains in mast cell α-granules has been reported [23]. Seglycin is a predominant proteoglycan that is modified with heparin or CS polysaccharides in mast cell α-granules. Its side chains are partially hydrolyzed by heparanase or HYAL4 to release GAG side chains and their binding proteases during degranulation [23,28,29,30]. Heparanase has been demonstrated to be secreted by cancer cells to promote their invasion and metastasis by degrading heparan sulfate in the extracellular matrix [31,32]. High expression of HYAL4 in some cancer tissues and cells has been reported [24]. Lokeshwar et al. demonstrated that a splice variant (V1) of *HYAL4* is upregulated in bladder cancer and drives a malignant phenotype [33]. Hyal4 may be involved in cancer invasion and metastasis by degrading CS-proteoglycans in the extracellular matrix. In addition, mutations in the *HYAL4* gene have been reported for some cancers including colorectal, stomach, lung, bladder, glioblastoma, leukemia, head/neck, ovarian, breast, and kidney cancers [24,34,35,36]. Maciej-Hulme roughly calculated that 4% of endometrial tumors have mutations in the *HYAL4* gene. It is necessary to investigate the mechanism linking HYAL4 to cancer [24].

Since overexpression of *Hyal4* resulted in a decrease in cellular CS, HYAL4 may be involved in cellular catabolism of CS in addition to the function of granule homeostasis in mast cell α-granules [23]. The mechanism of CS catabolism has not been well elucidated. CS polysaccharides are considered to be depolymerized into smaller oligosaccharides by endo-type hydrolase(s), and then, exo-type sulfatases as well as glycosidases act on them from the nonreducing ends one by one to release monosaccharides and inorganic sulfate ions [17]. Although the former endo-type enzymes have not been investigated, the latter exo-type enzymes involved in CS catabolism have been well characterized, because of the lysosomal diseases, mucopolysaccharidoses. In patients with the diseases, the enzyme activities of such exo-type enzymes are completely lost or subtly reduced by mutations in those genes. HYAL4 may be one of the endo-type hydrolases involved in cellular catabolism of CS.

Recently, we generated *Hyal4*-deficient mice using the CRISPR/Cas9 system, and its phenotype is under investigation. They are alive and fertile. Thus far, no clear anomalies have been detected. Since the predominant expression of *Hyal4* in the testis has been found, effects on male fertilization should be examined more precisely. The number of offspring produced by *Hyal4*-deficient mice will be compared statistically with that of wild-type mice. Accumulation of CS in several organs of the mice should be examined. Useful tools for HYAL4 analysis are limited thus far. *Hyal4*-deficient mice will be important to generate new information to understand the various functions of HYAL4 in vivo.

## 4. Materials and Methods

### 4.1. Animals and Ethics for Animal Experiments

BALB/c and C57BL/6J mice were obtained from Japan SLC Inc. (Hamamatsu, Japan). All experimental protocols including the use of laboratory animals were approved by the Animal Ethics Board of Meijo University (approval numbers: 2020-5, 2021-5, and 2022-5) and followed the guidelines of the Japanese Pharmacological Society (Folia Pharmacol. Japan, 1992, 99: 35A). CHO cells were obtained from the Japanese Collection of Research Bioresources (JCRB).

### 4.2. Reverse Transcription Polymerase Chain Reaction (RT-PCR)

To reevaluate the expression of Hyal4 mRNA in various tissues, RT-PCR was performed using newly designed primers. Total RNA was extracted from mouse tissues with a Trizol reagent (Thermo Fischer Scientific, Waltham, MA, USA), and cDNA was synthesized from 1 μg of total RNA using Moloney Murine Leukemia Virus reverse transcriptase (Promega) and an oligo(dT) primer [21]. The mouse *Hyal4* transcript was amplified from cDNAs of the multiple tissues or the commercial MTC Multiple Tissue Panels (Clontech) by PCR using the primers 5′-CAAGATCTAATTAGTACCATAGGAG-3′ and 5′-AATGGAGGTAATGAGCTGCTT-3′. Each PCR was carried out with the KOD Fx Neo DNA polymerase (Toyobo Co., Ltd., Osaka, Japan) for 40 cycles at 98 °C for 10 s, 54 °C for 30 s, and 68 °C for 30 s. PCR products were analyzed by 3% agarose gel electrophoresis.

### 4.3. Semiquantitative and Real-Time RT-PCR

Expression levels of *Hyal4* mRNA in various tissues were semiquantitatively analyzed. cDNAs of the multiple tissues or the commercial MTC Multiple Tissue Panels were subjected to semiquantitative and real-time PCR on LightCycler 480 Instrument using LightCycler 480 SYBR^®^ Green I Master (Roche Molecular Biochemicals, Mannheim, Germany). The amplification setting consisted of an initial denaturation at 95 °C for 5 min followed by 45 cycles of denaturation at 95 °C for 15 s, annealing at 50 °C for 20 s, and extension at 72 °C for 20 s. After amplification, melting-curve analysis was performed by heating the real-time PCR products at 95 °C for 5 s, and cooling at 65 °C for 60 s. To verify the specificity of each primer, melting-curve analysis was included. To determine the relative mRNA levels of the mouse *Hyal4* gene, the housekeeping gene, glyceraldehyde-3-phosphate dehydrogenase (*Gapdh*), was used as an internal control. The primer set for the *Gapdh* gene was as follows: mouse, forward, 5′-GATGACATCAAGAAGGTGGTGA-3′, and reverse, 5′-TGCTGTAGCCGTATTCATTGTC-3′.

### 4.4. Western Blotting

Expression levels of HYAL4 were examined not only at the mRNA level but also at the protein level. To detect mouse HYAL4 protein, Western blotting was performed. Samples extracted from various tissues of a C57BL/6J mouse or fractionated from L6 cells were subjected to SDS-PAGE using 10% SDS-polyacrylamide gels [20], and proteins were transferred to a PVDF membrane. The membrane was incubated with anti-HYAL4 antibody (A-7) (Santa Cruz Biotechnology, Inc., Heidelberg, Germany) diluted 1:1000 with 25 mM Tris-buffered saline containing 0.05% Tween 20 (TBST) overnight and then with ECL™ HRP-labeled anti-mouse IgG antibody (GE Healthcare) diluted 1:10,000 with TBST for detection of the samples prepared from cultured cells. An HRP-conjugated anti-HYAL4 antibody (A-7) (Santa Cruz Biotechnology, Inc.) was used as a primary antibody, when tissue samples were examined. The bound antibody was detected using an Immunostar LD Western blotting detection kit (Fujifilm, Osaka, Japan).

### 4.5. Flow Cytometry Analysis

To clarify whether the HYAL4 protein is present on the cell surface, flow cytometry analysis was performed using anti-HYAL4 antibody. Rat skeletal muscle myoblasts, L6 cells, were obtained from JCRB, National Institute of Biomedical Innovation (Osaka, Japan), and were cultured in Dulbecco’s modified Eagle’s medium (DMEM) containing 2.5% fetal calf serum under humidified conditions, and 5% CO_2_–95% air at 37 °C.

Single-cell suspension from L6 cells was obtained by mechanical disruption with 100 mM phosphate-buffered saline (pH 7.4) (PBS) and straining through a 40 mm nylon mesh. Cells were counted and surface-stained with the primary antibody, anti-HYAL4 antibody (clone A-7), and the secondary antibody, Alexa Fluor 488-conjugated anti-mouse IgG(H + L) antibody (Abcam Inc., Cambridge, UK), on ice for 1 h each. After washing with PBS containing 0.1% bovine serum albumin three times, the cells were analyzed using a BD LSRFortessa^TM^ X-20 flow cytometer (BD Biosciences, Franklin Lakes, NJ, USA).

### 4.6. Preparation of Membrane and Lysosome Fractions

Cellular localization of HYAL4 protein was investigated after fractionation of organelles followed by Western blotting. To prepare subcellular fractions from L6 cells, ProteoExtract^®^ Subcellular Proteome Extraction Kit (Merck., Kenilworth, NJ, USA) was used, which enables extraction of mammalian proteins from the cytosolic, organelle and membrane, nuclear, and cytoskeletal fractions, by following the manufacturer’s instructions. Each fraction was subjected to Western blotting as described above. The lysosome fraction was isolated and enriched using Thermo Scientific™ Lysosome Enrichment Kit (Thermo Fisher Scientific, Rockford, IL, USA) from L6 cells, according to the manufacturer’s instructions. The extracted fraction was analyzed by Western blotting using an anti-HYAL4 antibody (A-7) as well as an anti-LAMP1 (lysosome marker) antibody (Abcam) as a positive control.

### 4.7. Overexpression of Mouse HYAL4 in CHO Cells

To examine the effect of HYAL4 on the cellular catabolism of CS, the mouse *HYAL4* gene was overexpressed in CHO cells. Transfection was performed using exponentially growing cells at 70% confluency in a 9-cm^2^ culture dish. The cells were transfected with 5 µg of pEF6/V5-His expression vector (Invitrogen) containing the full-length *Hyal4* gene using Lipofectamine 3000 (Invitrogen). After selection with blasticidin S for 4 weeks, stably transfected cells were obtained. Disaccharides of CS and heparan sulfate in the cells were quantified after digestion with the CS- and heparan sulfate-specific eliminase, respectively, as described previously [37]. Recombinant DNA experiments were approved by the Biosafety Committee for Recombinant DNA Research of Meijo University (approval number: 2017-104).

## Figures and Tables

**Figure 1 molecules-27-06103-f001:**
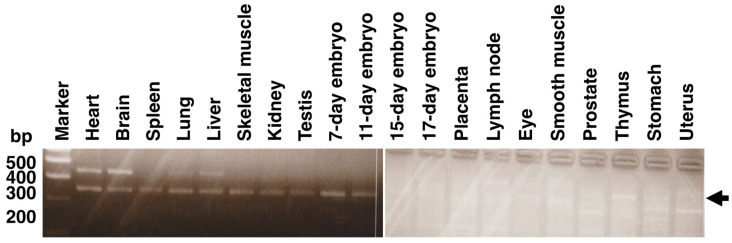
Tissue distribution analysis of mouse *Hyal4* mRNA. The expression pattern of mouse *Hyal4* mRNA was examined by PCR using cDNA from various tissues from a BALB/c mouse. Since the mouse *Hyal4*-specific primers were newly designed, the band at around 250 bp, indicated by an arrow on the right, was detected in all the tissues examined, demonstrating that it is ubiquitously expressed, in contrast to the previous studies. An extra band was detected at around 300 bp in the brain and spleen. It may have been amplified nonspecifically.

**Figure 2 molecules-27-06103-f002:**
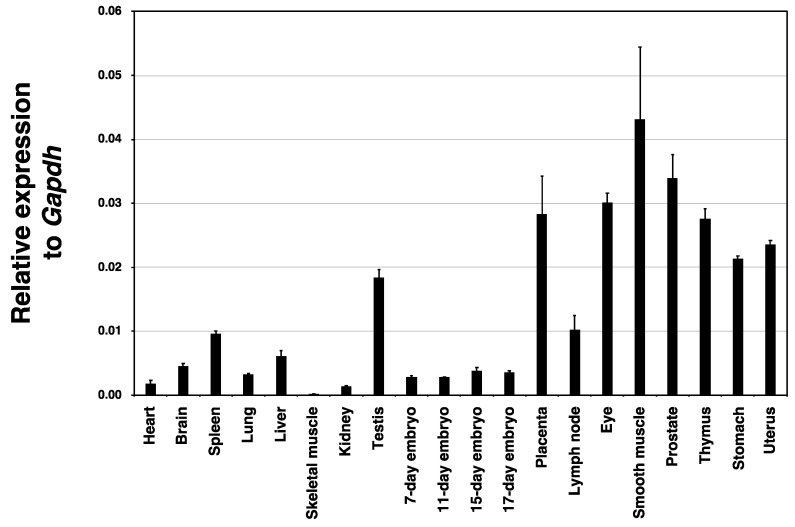
Semi-quantitative analysis of *Hyal4* mRNA in various mouse tissues by real-time RT-PCR. MTC Multiple Tissue Panels of a BALB/c mouse were used for the assay. The levels of mouse *Hyal4* were quantified by real-time RT–PCR using *Gapdh* as an internal standard. Values represent means ± standard error (n = 3).

**Figure 3 molecules-27-06103-f003:**
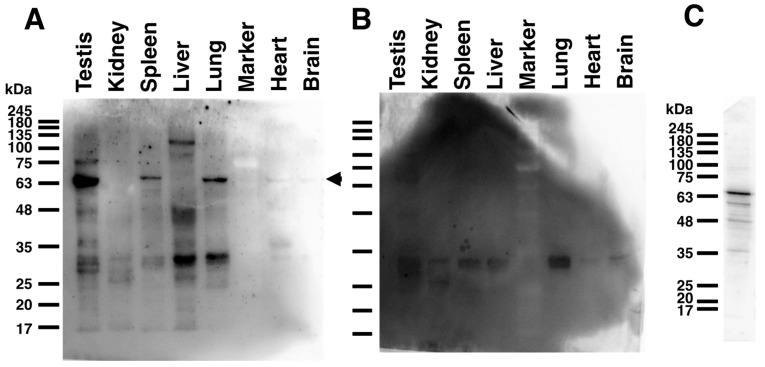
Tissue distribution analysis of mouse HYAL4 protein in various mouse tissues. Expression levels of mouse HYAL4 protein in various tissues of a C57BL/6J mouse were examined by Western blotting using an anti-HYAL4 antibody (A-7), HRP-conjugate (**A**). HRP-conjugated normal mouse IgG was used instead of the primary antibody for the control experiments (**B**). HYAL4 protein expressed in L6 cells was detected by the HRP-conjugated anti-HYAL4 antibody A-7 as a positive control (**C**). The band detected at around 70 kDa indicated by an arrowhead on the right may be a glycosylated form of mouse HYAL4 protein (*A*). Bands detected at around 35 kDa may be nonspecific binding, as also detected in the control panel (**B**).

**Figure 4 molecules-27-06103-f004:**
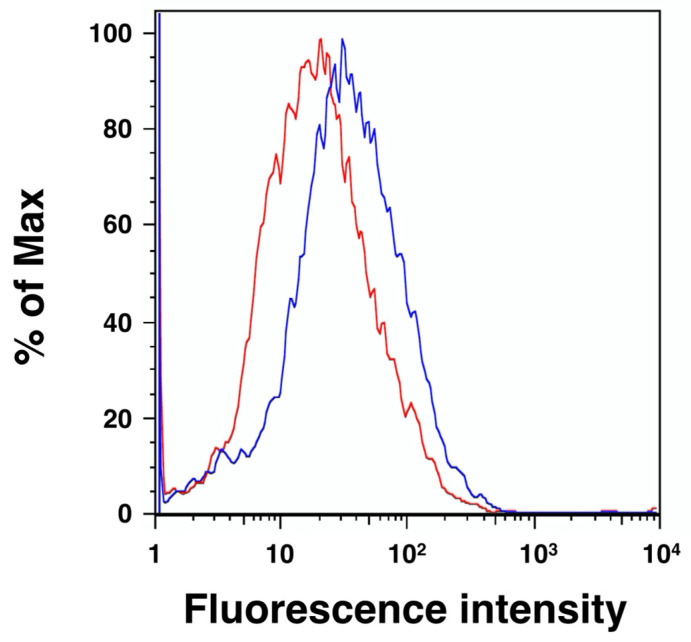
Flow cytometric analysis of the HYAL4 protein on the surface of L6 cells. Representative line graphs of cellular staining are shown. Staining of L6 cells was performed using anti-HYAL4 antibody (clone A-7) (blue) or normal mouse IgG (red) under nonpermeabilized conditions. The binding of these antibodies to the epitopes on the cell surface was visualized by flow cytometry after incubating with Alexa Fluor 488-conjugated secondary antibody. Increased expression of HYAL4 on the cell surface was noted.

**Figure 5 molecules-27-06103-f005:**
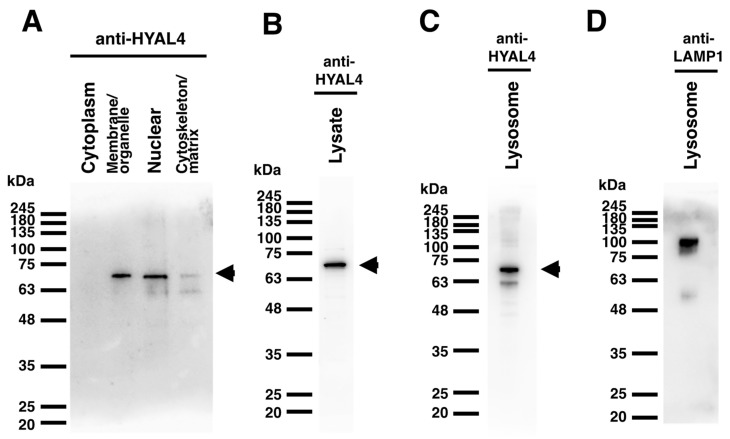
Panel (**A**): Proteins in L6 cells were extracted and separated into cytosolic, organelle and membrane, nuclear, and cytoskeletal fractions using Subcellular Proteome Extraction Kit. An aliquot of each fraction was subjected to SDS-PAGE and transferred to a polyvinylidene difluoride (PVDF) membrane for blotting with the HYAL4 antibody. Panel (**B**): Whole cell lysate of L6 cells was subjected to Western blotting analysis with the HYAL4 antibody. Panels (**C**,**D**): Proteins in lysosomes of L6 cells were enriched and extracted by Lysosome Enrichment Kit. An aliquot was subjected to SDS-PAGE and transferred to a PVDF membrane for blotting with the HYAL4 antibody (**C**) and LAMP1 antibody (**D**). The band detected at around 70 kDa indicated by arrowheads on the right (**A**–**C**) may be a glycosylated form of mouse HYAL4 protein. When the normal IgG from mouse serum was used as the primary antibody, no bands were detected (results not shown).

**Table 1 molecules-27-06103-t001:** Yields of CS disaccharides in CHO cells overexpressed with mouse HYAL4.

Disaccharide	Mock ^1^	*Hyal4* ^1^
ΔHexA-GalNAc ^2^	ND ^3^	ND
ΔHexA-GalNAc(4-*O*-sulfate)	224.4	77.2
ΔHexA-GalNAc(6-*O*-sulfate)	2.3	1.1
ΔHexA(2-*O*-sulfate)-GalNAc(6-*O*-sulfate)	ND	ND
ΔHexA-GalNAc(4,6-*O*-disulfate)	ND	ND
Total	226.7	78.3

^1^ The values are given as pmol/mg protein and represent the mean of those obtained from two independent experiments. ^2^ CS is composed of repeating disaccharide units of GlcA and GalNAc with sulfations at various positions. Digestion of CS with bacterial CS-lyase into disaccharide units converts GlcA into unsaturated hexuronic acid (ΔHexA). ^3^ ND, not detected.

## Data Availability

The data presented in this study are available in Appendix A.

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
