# Peer review of "Characterization of Hyaluronidase 4 Involved in the Catabolism of Chondroitin Sulfate"

_molecules, 2022, doi:10.3390/molecules27186103_

Round 1

Reviewer 1 Report

The main aim of this manuscript is to study the distribution of HYAL4 in organs and tissues and its cellular location. The authors conclude that various mouse organs expressed HYAL4; However, the study lacks positive and negative controls and the use of human tissues to corroborate that the expression of HYAL4 occurred in multiple tissues.

Mayor comments

1. Something went wrong with the upload of figure 1 as there are no bands showing up in this figure.

2. The antibody used to detect HYAL4 expression seems unspecific as multiple bands appeared in the western blot. Were other anti- HYAL4 antibodies tested?

3. Please verify the protocol used for HYAL4 western blot detection. In the result section, the authors mention that an HRP-linked anti- HYAL4 was used, but in the methods, it is described that the membrane was blot with an anti- HYAL4 antibody followed by an HRP anti-mouse antibody.

3. The use of recombinant mouse HYAL4 as a positive control in the western blot is missing, and the screeing of other tissues, such as the placenta and smooth muscle. Since the testis, placenta, smooth muscle, and other tissues, such as the prostate and thymus, present significantly higher levels of Hyal4 mRNA, the use of these tissues would be a good positive control, and the analysis of tissues from Hyal4-deficient mice would be a good negative control. What was the amount of protein used for the screening? Was the same amount of protein per tissue analyzed?

4. western blot of the whole L6 cell lysate is missing. Also, why does the band from the organelle and nuclear fractions appear at ~70 kDa while the band from the Lysosomes appears at ~ 63 kDa?

5. Please show evidence that the CHO transfected with the mouse Hyal4 gene overexpress the protein. Is the protein expressed on the cell membrane and in the lysosomes too?

Reviewer 2 Report

The manuscript entitled “Characterization of hyaluronidase involved in the catabolism of chondroitin sulfate” summarized the Hyaluronidase (HYAL) is an endo-type N-acetylhexosaminidase which plays an important role in the catabolism of hyaluronan as well as chondroitin sulfate.  Generally, the manuscript is very poor. This paper has several weaknesses and needs improvement before publication.

This manuscript has major language problems. There are too many for me to modify them all. Authors are strongly encouraged to seek a native English speaker who may assist you modifying the document.

Comments:

1.     Insert the scale in all figures.

2.     Summarize the abstract, focus on the main findings and mention the small conclusion in at the end of abstract

3.     In the Introduction focus on the objectives and insert a few new reference and relevant findings

4.     Material and method needs to clarifying and summarizing- some detailed needs

5.     The subtitles in the material and method needs to summarizing Ethical approval and references must be mentioned in M&M

In conclusion, the research presented is interesting, well planned and carried out. The manuscript can still be improve revise by a native English speaker. Nevertheless, I believe that this work deserves publication in after the inclusion of corrections.

Round 2

Reviewer 1 Report

 The authors addressed all the reviewer's concerns. I do not have any further comments.

Reviewer 2 Report

I appreciate your efforts to respond to my review comments.